# MindGPT: Interpreting What You See with Non-invasive Brain Recordings

## Abstract

Decoding of seen visual contents with non-invasive brain recordings has important scientific and practical values. Efforts have been made to recover the seen images from brain signals. However, most existing approaches cannot faithfully reflect the visual contents due to insufficient image quality or semantic mismatches. Compared with reconstructing pixel-level visual images, speaking is a more efficient and effective way to explain visual information. Here we introduce a non-invasive neural decoder, termed MindGPT, which interprets perceived visual stimuli into natural languages from fMRI signals in an end-to-end manner. Specifically, our model builds upon a visually guided neural encoder with a cross-attention mechanism. By the collaborative use of data augmentation techniques, this architecture permits us to guide latent neural representations towards a desired language semantic direction in a self-supervised fashion. Through doing so, we found that the neural representations of the MindGPT are explainable, which can be used to evaluate the contributions of visual properties to language semantics. Our experiments show that the generated word sequences truthfully represented the visual information (with essential details) conveyed in the seen stimuli. The results also suggested that with respect to language decoding tasks, the higher visual cortex (HVC) is more semantically informative than the lower visual cortex (LVC), and using only the HVC can recover most of the semantic information.

## 1 Introduction

Humans can describe the visual objects of the world using a finite number of words, and draw an analogy between verbal and visual when communicating with others. This flexible cognition capacity suggests that semantic information, conveyed in language, is deeply intertwined and entangled with various types of sensory input, especially for vision. Neuroscience studies (Popham et al., 2021; Tang et al., 2023; Fairhall & Caramazza, 2013; Binder & Desai, 2011) hold that amodal semantic representations are shared between visual and linguistic (V&L) perceptions, e.g., the word "cat" evokes similar conceptual content to the image of a cat in our mind. However, how the brain infers semantic relations of conceptual categories, and fulfills seamless switching between V&L modalities has been rarely quantized or implemented with computational models.

Recent neural decoders (Chen et al., 2023a;b; Takagi & Nishimoto, 2023) demonstrated that visual content can be reconstructed from visual cortex (VC) representations recorded using functional Magnetic Resonance Imaging (fMRI). Nevertheless, the reconstructed images still suffered from being blurry and semantically meaningless or mismatched. For another, the neuroscience community has presented compelling evidence (Popham et al., 2021) to support the notion that semantic concepts in both V&L forms can be accessed in the brain's VC. The findings strongly encourage us to introduce a new "mind reading" technology, aiming to verbally interpret what you see. Such an endeavor has great scientific significance in revealing cross-modal semantic integration mechanisms and may provide potential application values for restorative or augmentative BCIs.

Here, we introduce a non-invasive neural language decoder, termed MindGPT, which translates the blood-oxygen-level-dependent (BOLD) patterns elicited by static visual stimuli into well-formed word sequences, as shown in Fig. 1 Left. For the non-invasive language decoder, to the best of our knowledge, Tang et al. (2023) made the pioneering attempt to develop a non-invasive neural decoder for perceived speech reconstruction, which can even recover the meaning of silent videos. Due to

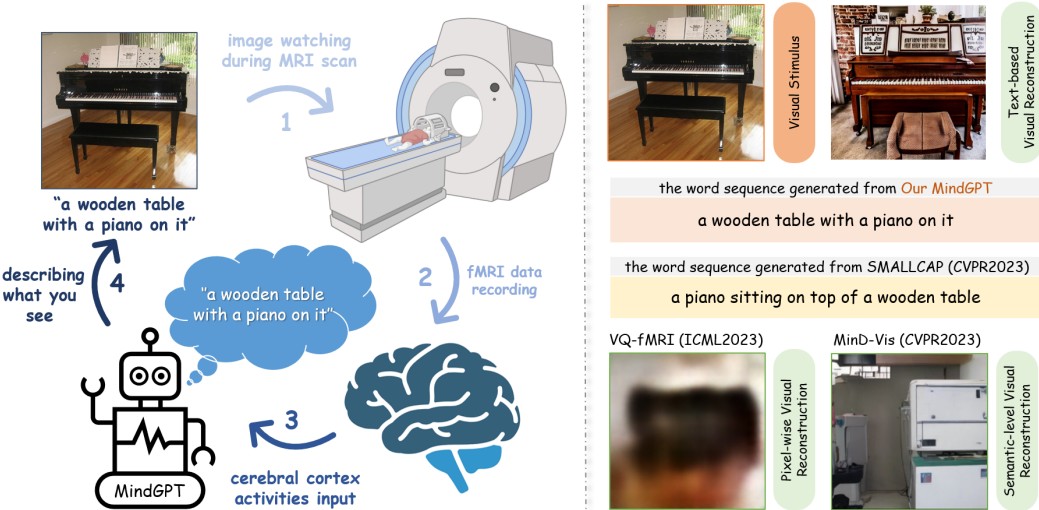

Figure 1: Left: The overall pipeline of non-invasive language decoder MindGPT. Right: Reconstruction results of our MindGPT, image captioning model SMALLCAP (Ramos et al., 2023), and visual decoding methods VQ-fMRI (Chen et al., 2023a) & MinD-Vis (Chen et al., 2023b).

the poor temporal resolution of fMRI, however, the method requires collecting a large amount of fMRI signals (recorded while subjects listened to spoken stories) to predict the fine-grained semantic relevance between the candidate words and the evoked brain responses. On the contrary, this study focuses on whether and to what extent the static visual sensory experiences such as a single image provide semantic labels for our amodal language maps.

Our MindGPT must meet two key criteria: i) the capability of capturing visual semantic representations (VSRs) from brain activities, and ii) the incorporation of a mechanism to transition from acquired VSRs into well-formed word sequences. To do so, firstly, we opt to employ a large language model GPT-2 (Radford et al., 2019), as our text generator, thus allowing us to constrain sentence structures to resemble well-formed natural language. We then customize a simple yet efficient CLIP-guided (Radford et al., 2021) fMRI encoder with cross-attention layers to bridge the semantic gap between brain-visual-linguistic (B&V&L) representations in an end-to-end fashion. Finally, by using pseudo-labels, we present a data augmentation technique to construct biologically meaningful supervision signals from limited annotations. This formulation, unlike previous works that rely on linear models (Mai & Zhang, 2023), permits us to explore self-supervised neural semantics learners.

In this study, we have demonstrated that the MindGPT could be the bridge of robust V&L semantic transformations of the brain's VC and machine. The language generated by our MindGPT reflects the visual semantics of the observed stimuli (see Fig. 1 Right) with high accuracy, which suggested that our method successfully learned the generalizable neural semantic representations, and gained a wide understanding of B&V&L modalities. Furthermore, we found that the well-trained MindGPT appears to emerge with the ability to capture visual cues (i.e., salient regions) of stimulus images, even from highly limited fMRI-image training data, which facilitates us to explore the contributions of visual properties to language semantics. With the help of visualization tool, we also observed that the latent neural representations learned by MindGPT exhibited desirable locality-sensitive properties both in low-level visual features and high-level semantic concepts, which conforms to some neuroscience findings (Bellmund et al., 2018; Yamins & DiCarlo, 2016). Overall, our MindGPT, different from Tang et al. (2023), indicated that the semantic relations between V&L representations can be inferred from our brain's VC without consideration for temporal resolution of fMRI.

## 2 RELATED WORK

The neural decoding technique offers a unique fashion for advancing our understanding of human perception. With deep learning technological changes (Goodfellow et al., 2014; Radford et al.,

2021; Kingma & Welling, 2013; Ho et al., 2020; Rombach et al., 2022) and neuroscience advances (Haxby et al., 2001; Kamitani & Tong, 2005; Yamins & DiCarlo, 2016; Popham et al., 2021), the visual neural decoding community is progressing quickly. In recent decades, a lot of inspiring work with vital guiding implications has sprung up, which can be broadly broken down into three main paradigms based on decoding objectives (Du et al., 2023), i.e., stimuli classification (Haxby et al., 2001; Van Gerven et al., 2010; Damarla & Just, 2013; Yargholi & Hossein-Zadeh, 2016; Du et al., 2023), recognition (Haynes & Rees, 2006; Kay et al., 2008; Horikawa & Kamitani, 2017; Naselaris et al., 2009), and reconstruction (Beliy et al., 2019; Lin et al., 2022; Chen et al., 2023a;b;c). Among them, visual reconstruction, which aims to recover the overall organization of seen images, is the most challenging yet exciting. In the remaining section, we will briefly review the background material of reconstruction tasks, that puts our study into context.

The key to the success of image reconstruction techniques is to extract low-level image details of visual stimuli from brain activity using fMRI. Interestingly, for the target of visual reconstruction tasks, there has been a trend in recent years away from pixel-wise reconstruction and toward seeking the semantically correct images (namely, allowing visual structure variance under the same semantics) with the rise of diffusion models (Ho et al., 2020; Rombach et al., 2022). The decoded outcomes of early techniques (Shen et al., 2019b;a; Beliy et al., 2019; Ren et al., 2021; Du et al., 2022) can preserve the outlines and postures of original stimuli, but they often fail to recover the intricate texture and rich color in natural scenes due to the limited number of fMRI-image annotations. On the other hand, high-level semantic decoding methods incorporate visual semantic information into the GAN models (Mozafari et al., 2020; Ozcelik et al., 2022) or diffusion models (Lu et al., 2023; Takagi & Nishimoto, 2023; Chen et al., 2023b;c), resulting in realistic images due to inherited strong generative capabilities. However, the models lack control over low-level details such as contour and texture. More importantly, the reconstructed image usually has a large semantic gap with the actual stimulus, leaving it difficult to interpret what you see. For humans, remembering the detail of a seen scene is a tricky issue since our visual system is not like a camera that stores every pixel of images (Chen et al., 2023a; Desimone et al., 1995), but we are skilled at a general description of the seen objects, meaning that speaking is a simple but more effective fashion of presenting visual semantics. Previous works (Matsuo et al., 2018; Takada et al., 2020; Mai & Zhang, 2023; Ferrante et al., 2023) mapped fMRI recordings to the embeddings of pre-trained neural networks like VGGNet by relatively simple linear regression models, and then feeds the predicted result into language models to generate word sequences. Unlike existing decoding paradigms, our MindGPT is designed to explore self-supervised neural semantics reconstruction by using cross-attention mechanisms and data augmentation techniques. To the best of our knowledge, generating linguistic semantic information directly from a single brain image in an end-to-end fashion has not been adequately explored.

## 3 THE MINDGPT APPROACH

MindGPT is a lightweight non-invasive neural decoder, which combines off-the-shelf large language model GPT-2 (Radford et al., 2019) and pre-trained CLIP (Radford et al., 2021), to describe the meaning of perceived images by natural language, as shown in Fig. 2.

### 3.1 DATASET AND PREPROCESSING

In this study, a widely used benchmark dataset that was designed for fMRI-based decoding, termed as DIR (Shen et al., 2019b), was leveraged to evaluate our MindGPT. In natural image presentation experiments, including training and test sessions, three healthy subjects were required to view natural images selected from ImageNet (Deng et al., 2009), and simultaneously fMRI signals were collected using a 3.0-Tesla Siemens MAGNETOM Verio scanner. Each scanning session includes anatomical (inplane T2) and functional (EPI) images covering the entire brain (TR, 2 s; TE, 43 ms; voxel size, $2 \times 2 \times 2$ mm; number of slices, 76). The visual stimuli (1200 training images, and 50 test images) involved in the experiment are identical to those used in another fMRI-image dataset (Horikawa & Kamitani, 2017), but the DIR dataset contains a larger number of image-fMRI pairs ($5 \times 1200$ training samples, and $24 \times 50$ test samples). Note that 5 and 24 represent the number of repetitions. To avoid scanner instability effects, for each run, the first 8 s of scans were discarded. All fMRI data were subjected to 3-dimensional motion correction using SPM, and then co-registered to the high-resolution anatomical images and regions-of-interest (ROIs) selection (Shen et al., 2019b).

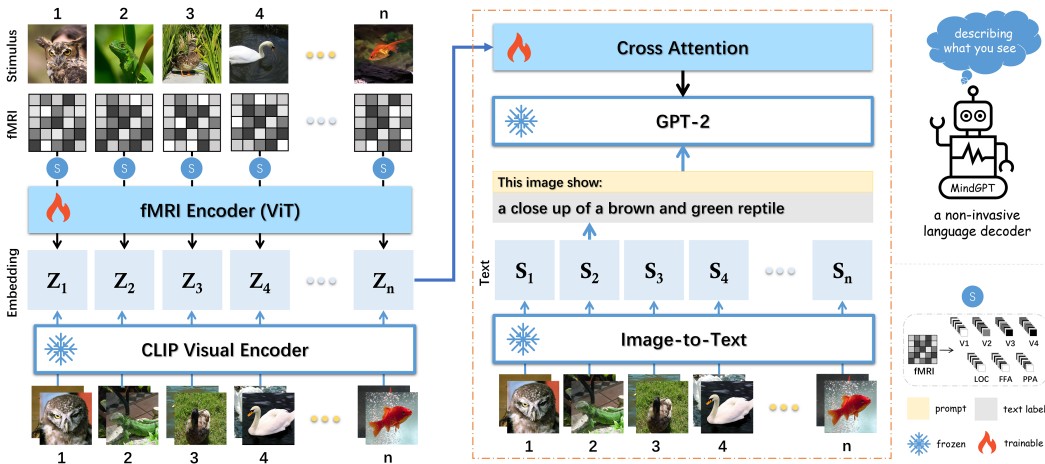

Figure 2: Schematic diagram of MindGPT framework. We first split an fMRI signal into fixed-size low-to-high-level ROIs (namely, V1-V4, LOC, FFA, and PPA), and feed the resulting sequence of voxel vectors to a standard ViT for fMRI visual representations learning guided by CLIP visual encoder. Then, we use trainable cross-attention modules to bridge a frozen GPT-2 and fMRI encoder. In this way, our model can generate a word sequence conditioned on the input fMRI.

In this study, we used the voxels from the brain's visual areas including V1-V4, LOC, FFA, and PPA, where V1 to V3 is defined as the lower visual cortex (LVC), and the higher visual cortex (HVC) is formed by LOC, FFA, and PPA (Horikawa & Kamitani, 2017).

## 3.2 CLIP-GUIDED NEURAL EMBEDDING

The goal of our MindGPT is the process of generating a descriptive sentence for brain activity patterns evoked by visual objects. To this end, the key here is to guide our model towards a desired visual direction (i.e., semantic information of stimulus images) with each generation step. Firstly, to handle fMRI signals, we split the fMRI into a sequence of voxel vectors $z \in \mathbb{R}^{7 \times H}$ including V1-V4, LOC, FFA, and PPA, where $H$ denotes the number of voxels, which is flattened and padded to the same size. Next, voxel vectors $z \in \mathbb{R}^{7 \times H}$ are fed into a trainable linear projection, followed by a Transformer encoder, to predict latent fMRI representations $\mathcal{Z}$. During the training phase, we leverage the hidden class embedding $\mathcal{K}_{clip} \in \mathbb{R}^{768}$ of CLIP visual encoder (Radford et al., 2021) as neural proxy, and then seeking a joint semantic space across images and fMRI signals via fMRI-image representation alignment. Moreover, since the size of the carefully curated dataset is fairly limited, we present a simple data augmentation strategy, building virtual training examples by performing linear interpolation on the fMRIs evoked by the same category of images. This practice shares similarities with mixup technique (Zhang et al., 2018), but the difference is that the corresponding labels are randomly sampled from the subset (annotated with the same category) of ImageNet (Deng et al., 2009) rather than generated via equal-weighted interpolation. By doing so, the model can be encouraged to extract shared high-level semantic features of augmented images.

## 3.3 VISION-LANGUAGE JOINT MODELLING

In order to restrict the decoded word sequences to well-formed language, our approach uses an autoregressive language model GPT-2 (Radford et al., 2019), which specializes in modelling text semantic interactions between the next token $s_i$ and past tokens $(s_1, s_2, \cdots, s_{i-1})$ at each time-step. Given any initial prompt, such as "The seen image shows", GPT-2 will infer the likelihood of words $P(s_i|[s_j]_{j<i})$ that could come next. Nevertheless, even with the constraints imposed by the prior probability distribution $P(S) = \prod_{i=1}^{n} P(s_i|[s_j]_{j<i})$ learned from WebText dataset (Radford et al., 2019), it may be computationally problematic to formalize visually-guided neural language decoding problem as $P(s_i|[s_j]_{j<i}, \mathcal{Z})$ directly. This is due to that the fMRI encoder and GPT-2 model operate in different embedding spaces.

For coupling the V&L representations, we use multi-head cross-attention layers to bridge the fMRI encoder and GPT decoder, thus leaving each layer of the GPT decoder attends to the fMRI encoder outputs (Vaswani et al., 2017). Under the circumstances, our task can be boiled down to an end-to-end multi-task optimization problem. Given an fMRI-image pair $(z, y)$, our loss function $\mathcal{L}_{mind}$ can then be written as

$$\mathcal{L}_{mind} = \mathcal{L}_{gpt}\Big(\mathbf{F}_t(y), \mathbf{E}_\Phi(z); \Theta\Big) + \mathcal{L}_{clip}\Big(\mathbf{E}_c(y), \mathbf{E}_\Phi(z)\Big), \qquad (1)$$

where $\mathbf{F}_t(y) = [s_i]_{1:M}$ is a visual captioning of image $y$ generated from SMALLCAP (Ramos et al., 2023), $\mathbf{E}_c(\cdot)$ denotes frozen CLIP encoder, which returns the hidden visual embedding $\mathcal{K}_{clip} \in \mathbb{R}^{768}$, $\mathbf{E}_\Phi(\cdot)$ indicates fMRI encoder with trainable parameters $\Phi$, and $\Theta$ is the weights in the cross-attention modules. The first term uses the standard cross-entropy loss for minimizing the sum of the negative log-likelihood conditioned on the fMRI embedding and the previous tokens, i.e.,

$$\mathcal{L}_{gpt} = -\sum_{i=1}^{M} log P(s_i | s_{<i}, \mathbf{E}_\Phi(z); \Theta). \qquad (2)$$

Note that we freeze the GPT decoder and CLIP encoder, and only train the randomly-initialized fMRI encoder as well as cross-attention layers. The second term of Eq. 1 is a mean-squared loss for alignment purposes:

$$\mathcal{L}_{clip} = \lambda \Big|\Big|[\mathbf{E}_c(y)]_0 - [\mathbf{E}_\Phi(z)]_0\Big|\Big|_2^2, \qquad (3)$$

where $[\cdot]_0$ returns the class embedding of Transformer encoder, and $\lambda = 10$ is a trade-off hyperparameter weighing $\mathcal{L}_{gpt}$ and $\mathcal{L}_{clip}$. Overall, our MindGPT provides a mechanism to learn a direct mapping between brain activity and text by preserving language attributes under the guidance of visual cues, which brings desirable expandability, i.e., our framework can easily be extended to other types of neural decoding such as fMRI-to-sound by an appropriate choice of the decoder. Moreover, as the result of avoiding separate visual feature decoding step, learning in an end-to-end fashion can effectively help in reducing the information loss (Shen et al., 2019a).

## 4 EXPERIMENTAL RESULTS

### 4.1 IMPLEMENTATION DETAILS AND EVALUATION METRICS

In this work, the architecture of our MindGPT contains two frozen pre-trained sub-models, CLIP-ViT-B/32 and GPT-2$_{\text{Base}}$, which are provided on HuggingFace (Wolf et al., 2020). In the MindGPT model, only the parameters of the fMRI encoder and cross-attention layers are trainable. For the fMRI encoder, we use a standard ViT model with an embedding size of 768, layer number of 8, and 8-head self-attention. The cross-attention layer with 12-head is added to each of the 12 layers of GPT-2 decoder. In order to further reduce the number of learnable parameters, following Ramos et al. (2023), we diminish the default dimensional size (64) of the projection matrices in the cross-attention layers to 8. During the training phase, we optimize MindGPT by using Adam solver (Kingma & Ba, 2014) with $\beta_1 = 0.9$, $\beta_2 = 0.999$, learning rate of 1e-4, and applying a low weight decay of 1e-4 until the model converges, which we found to be useful. Our MindGPT trained on DIR and a subset of ImageNet (Deng et al., 2009), including 150 categories totaling 200.7k images. Note that there's no overlap between the training and test categories. The MindGPT is implemented by Pytorch, and ran on 4 NVIDIA GeForce RTX3090 GPUs.

To provide an across-the-board evaluation of MindGPT's language decoding performance, we consider the following standard metrics: BLEU-1 (B@1), BLEU-4 (B@4) (Papineni et al., 2002), ROUGE-L (Lin & Hovy, 2003), METEOR (Denkowski & Lavie, 2014), CIDEr (Vedantam et al., 2015) and SPICE (Anderson et al., 2016), which widely used in various NLP tasks, e.g., translation, and image-to-text (image captioning) (Tewel et al., 2022; Ramos et al., 2023). These language similarity metrics are calculated using COCO evaluation package.

### 4.2 NEURAL DECODING ACROSS VISION AND LANGUAGE

**Qualitative Results.** In order to provide an intuitive understanding of the linguistic decoding capacity guided by visual stimuli, Fig. 3 reports few-shot and zero-shot generation examples from subject 3 of the DIR dataset. Note that the default training/test split of DIR has no overlapping image categories, we randomly sampled 50 fMRI-image training pairs, and added them to the test set for the

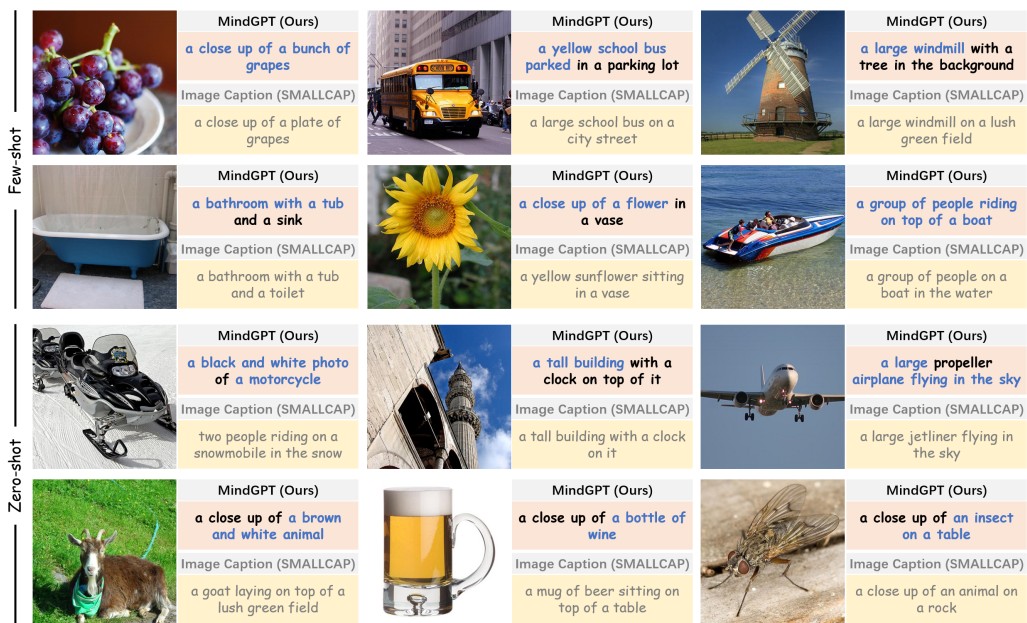

Figure 3: The language decoding results of our MindGPT. Top: Reconstruction results on known visual categories. Bottom: Reconstruction results on unknown visual categories that are out of the training set (zero-shot). For each group, the left represents the raw visual stimuli, the right reports the neural language decoding results of our MindGPT and image captioning results of SMALLCAP. Correct (or semantically similar) captions are highlighted in blue.

few-shot evaluation. For each group of results, the right shows the linguistic decoding result of our MindGPT, and provides the reference caption generated by Ramos et al. (2023). From the results, we see that MindGPT can produce semantically satisfying word sequences in both few-shot and zero-shot decoding, which extracted not only the meaning of the raw visual stimuli but often even exact category names such as "airplane", "windmill", "grapes", "school bus", and "bathroom". This demonstrates that fine-grained language semantics information can be recovered from the BOLD signal evoked by visual objects. Interestingly enough, we observe that our MindGPT appears to exhibit the capability to capture color information or infer the color tones of images, e.g., "black and white photo" (col 1, row 3), "brown and white animal" (col 1, row 4), "yellow school bus" (col 2, row 1). Moreover, although our method may not consistently infer correct classes of objects, it can still decode approximate semantic information, e.g., "beer"–"wine" (col 2, row 4), "fly"–"insect" (col 3, row 4), and "sunflower"–"flower" (col 2, row 2), which supports the assumption that V&L semantic information are well-represented in visual cortex (Popham et al., 2021).

**Quantitative Results.** Here, we report quantitative results of our MindGPT on different model configurations. For convenience, we use brief notation to indicate the model variants. For example, compared to the base model, MindGPT-S/8 means the smaller variant, and the scaling factor $N = 8$ of cross-attention layers. Note that the number of parameters is inversely proportional to the scaling factor $N$. The results, as summarized in Tab. 1, are based on the subject 3 of the DIR. From the Tab. 1, a few patterns can be observed. Firstly, the larger model MindGPT-L outperforms MindGPT-B and MindGPT-S on a range of language similarity metrics. Specifically, with the BLEU-4, which reflects the matching precision of four consecutive words (i.e., 4-gram), the MindGPT-L/16 is 21% to 27% higher than the MindGPT-B and MindGPT-S. With the ROUGE, which is mainly designed to consider recall rate, the MindGPT-L/16 obtains a high value of 41.7. For the CIDEr, which calculated the semantic similarity between sentences and used TF-IDF to consider word frequency, performance peaked at 116.5 with MindGPT-L/16 and decreased as the parameters of cross-attention layers increased. Under the SPICE, which computes the semantic matching degree between generated descriptions and the reference texts, the larger model, MindGPT-L/16 achieves a high value of 15.2, which is 29% to 52% higher than the other model variants. Secondly, we also note that decoding performance not only depends on the size of the fMRI encoder, but also on the cross-attention

layers. The reconstruction quality generally increased as cross-attention parameters decreased, i.e., the smaller cross-attention modules are good for performance, which is somewhat surprising. Our MindGPT may have not reached saturation yet within the range tried, we leave it to future work.

| Model | fMRI Encoder | | Cross-Attention | Params | Language Similarity Metrics ↑ | | | | | |
|---|---|---|---|---|---|---|---|---|---|---|
| | Layers | Heads | | | B@1 | B@4 | ROUGE-L | METEOR | CIDEr | SPICE |
| MindGPT-S/4 | | | N = 4 | 38M | 34.1 | 10.7 | 32.6 | 10.5 | 39.2 | 7.2 |
| MindGPT-S/8 | 4 | 4 | N = 8 | 35M | 37.9 | 15.9 | 36.4 | 12.9 | 65.7 | 10.0 |
| MindGPT-S/16 | | | N = 16 | 33M | 37.5 | 17.0 | 36.9 | 12.9 | 89.6 | 10.0 |
| MindGPT-B/4 | | | N = 4 | 67M | 38.8 | 15.4 | 37.0 | 13.1 | 64.0 | 10.4 |
| MindGPT-B/8 | 8 | 8 | N = 8 | 63M | 37.9 | 15.7 | 35.9 | 12.8 | 70.8 | 10.3 |
| MindGPT-B/16 | | | N = 16 | 61M | 39.7 | 16.2 | 39.2 | 13.8 | 77.3 | 11.8 |
| MindGPT-L/4 | | | N = 4 | 123M | 35.7 | 11.5 | 34.7 | 11.3 | 55.1 | 9.5 |
| MindGPT-L/8 | 16 | 16 | N = 8 | 120M | 40.8 | 17.5 | 40.4 | 14.4 | 75.2 | 12.3 |
| MindGPT-L/16 | | | N = 16 | 118M | **42.1** | **20.5** | **41.7** | **15.5** | **116.5** | **15.2** |

Table 1: Quantitative results of neural language reconstruction. We report the decoding performance of our MindGPT on the DIR default test set. Note that all training parameters are set to the default for different model configurations. The **best** and worst are highlighted in **bold** and red, respectively.

## 4.3 THE IMPACT OF HIERARCHICAL CODING PROPERTY ON LANGUAGE RECONSTRUCTION

In neuroscience, a fairly well-accepted theory is that visual information propagation from the lower visual cortex (LVC) to the higher visual cortex (HVC) has a hierarchical nature (Yamins & DiCarlo, 2016; Horikawa & Kamitani, 2017). This finding has been widely studied in visual reconstruction tasks (Fang et al., 2020; Takagi & Nishimoto, 2023). However, it is unclear how the hierarchical structure of information affects our decoding at the granularity of words and phrases, which regions are consistently engaged in language reconstruction. In other words, are the LVC and the HVC complementary or redundant for language representations?

| Model | ROI Variants | Voxel Number | Language Similarity Metrics ↑ | | | | | |
|---|---|---|---|---|---|---|---|---|
| | | | B@1 | B@4 | ROUGE-L | METEOR | CIDEr | SPICE |
| | LVC (V1 + V2 + V3) | 6550 | 39.9 | 14.1 | 38.6 | 12.7 | 54.1 | 9.4 |
| MindGPT-B/8 | HVC (LOC + FFA + PPA) | 5633 | **40.8** | **17.8** | **39.4** | **14.6** | **91.4** | **13.0** |
| | VC (V4 + LVC + HVC) | 14034 | 37.9 | 15.7 | 35.9 | 12.8 | 70.8 | 10.3 |

Table 2: Language semantics predictions of different brain areas for perceived visual images. All results are computed by language similarity metrics between the MindGPT predictions and the corresponding image captions. The **best** and worst are highlighted in **bold** and red, respectively.

**Performance of Different Brain Areas.** To preliminarily validate the underlying contributions of different brain regions to the language decoding task, we repeatedly run quantitative experiments using fMRI voxels of different visual areas (VC, LVC and HVC). Here, voxels of LVC are composed of V1, V2, and V3, voxels from FFA, PPA, and LOC form the HVC, and VC denotes the whole visual cortex. It should be noted that the default model configuration MindGPT-B/8, and the same training strategy are used for all three experiments. Tab. 2 shows the results. We find two phenomena worth exploring: (1) Decoding from the HVC yielded the best performance on all language evaluation metrics; (2) the decoding performance of using complete VC is better than that of LVC. These evidences seem to point in that there is no complementary relationship between LVC and HVC. Does this mean that LVC is redundant in decoding tasks? For answers, we will perform the analysis studies of the latent neural representations in the next sub-section.

**Analysis of the Latent Neural Representations.** Our MindGPT model allows us to decode linguistic semantic information, in which the latent fMRI representations play a crucial role. Therefore, examining the representation distributions of different brain regions is beneficial to further explain the above phenomena. The dimension of latent representations is too big, so we leverage the t-SNE technique (Van der Maaten & Hinton, 2008), which can preserve the local structure of data in low-dimensional embedding space, to visualize the distributions of fMRI representations. We separately map VC, LVC, and HVC neural representations to 2-dimensional t-SNE embedding spaces, and put

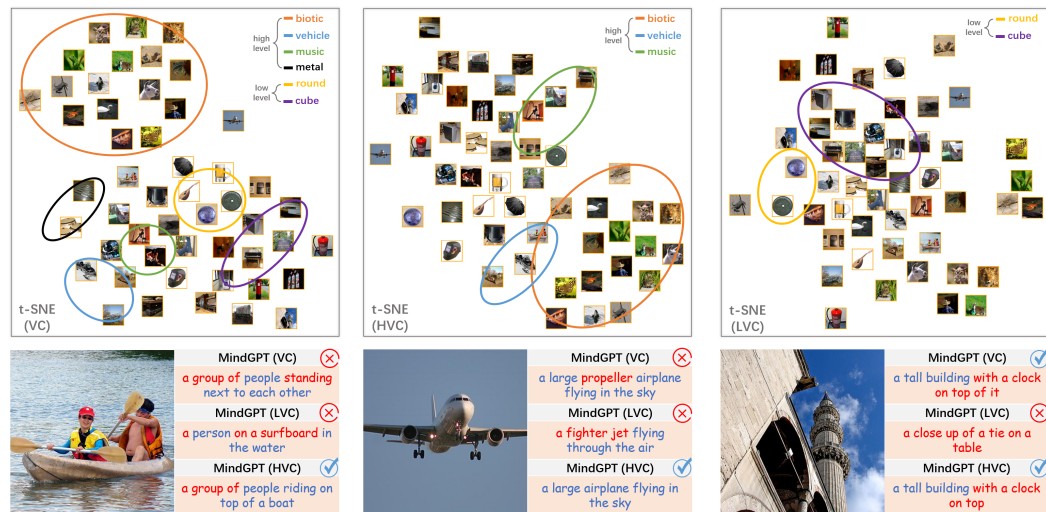

Figure 4: Top: The t-SNE visualization of neural representation for different brain areas. Bottom: Examples of our textual reconstruction conditioned on different brain regions (Red = false captions).

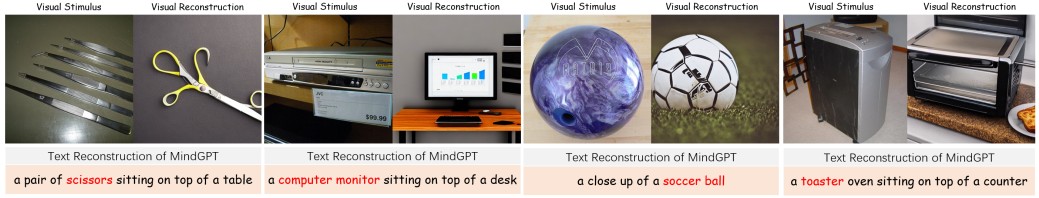

Figure 5: Typical imperfect cases of our MindGPT in textual reconstruction (Red = false captions). To intuitively understand semantic bias, we also provide visual reconstruction results obtained using the decoded text of our MindGPT and off-the-shelf Stable Diffusion (Rombach et al., 2022).

the corresponding visual stimulus at the position, as shown in Fig. 4 Top. From the visualization results of VC and HVC, we can observe that our MindGPT learned a locality-sensitive embedding structure, which contains several clusters representing different high-level semantic properties or concepts, e.g., biotic, vehicle, and music. The embedding structure of LVC, by contrast, has no obvious clustering rule. However, we can still find that similar low-level appearance features are located at nearby positions such as round and cube. In terms of the latent embedding space of VC, it inherits the semantic properties of low and high levels from LVC and HVC, but why is there a performance degradation when using the entire VC? The reason for the performance decline with VC may be that each brain region has a non-trivial probability of decoding failure, which means that the more brain areas we use, the harder it is to guarantee that all of the brain areas are always functional within the existing learning paradigm. We can see in Fig. 4 Bottom that low-level visual features are usually insufficient for effective semantic reconstruction, which tend to generate semantically inaccurate targets that are similar in appearance. More failure examples are provided in Fig. 5. To more intuitively evaluate the semantic reconstruction deviation, on the right of each example, we use off-the-shelf Stable Diffusion (version 1.4) (Rombach et al., 2022) with PLMS sampler to reconstruction visual stimuli (without fine-tuning) by conditioning on our linguistic decoding results.

## 4.4 DISCOVERING THE VISUAL CUES THAT GUIDE SEMANTIC RECONSTRUCTION

At present little is known about how the MindGPT encodes or infers semantic relations between V&L representations. We question whether the muted success of MindGPT in linguistic decoding can be attributed to the appropriate modeling of visual cues. This is also in line with the characteristics of our human vision system: only a certain part of the rich visual information contained in an image that interests us is perceived by our brain (Chen et al., 2023a).

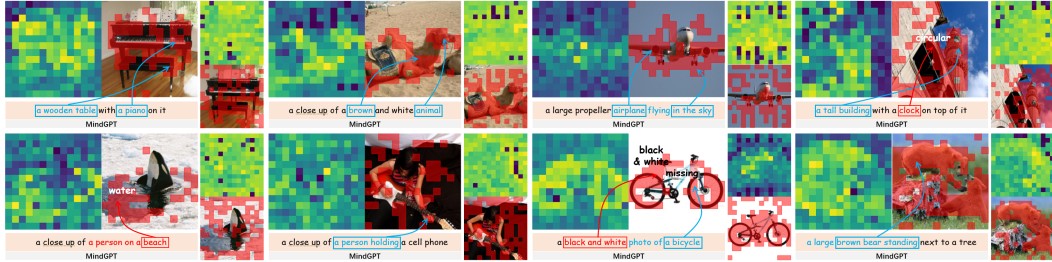

Figure 6: Schematic illustration of the semantic reconstruction guided by visual cues. On left of each group, we show the attention map based on the cosine similarity between fMRI and CLIP patch embedding, and its masking counterpart obtained by thresholding, respectively. The results derived from CLIP [CLS] and patch embeddings are also displayed on the right.

Typically, the [CLS] token's self-attention weighting coefficient of the ViT can be used to answer what a visual model is focusing on. However, the self-attention maps of our MindGPT encoder represent dependencies between different brain regions. In order to discover the visual cues that guide semantic reconstruction, our practice is using a CLIP visual encoder with $16 \times 16$ input patch size (i.e., CLIP-ViT-B/16) to produce a sequence of image patch embeddings, and then calculating the cosine similarity matrix (CSM) between each image patch embedding and the class embedding of fMRI. As shown qualitatively in Fig. 6, the CSMs contain information about the salient regions of an image. Note that we do not provide any supervision signals of salient positions in the form of labeled data or constraints during the training phase. We observe in Fig. 6 that the semantic reconstruction process is guided by attention-like visual cues, i.e., the masks of similarity maps are highly related semantically to the meaning of words or phrases in decoded language such as "a piano", "airplane flying in the sky", and "a tall building". The semantic deviation of reconstruction even can be explained by the visual cues. Specifically, for the $5^{th}$ example in Fig. 6, we can clearly see that fMRI representation focused on the water around a whale, thus decoding the word "beach". In the $6^{th}$ example, only the gesture of holding is captured, resulting in the decoded phrase "a person holding". As for the $7^{th}$ example, the mask nearly covers the key part of the bicycle, except for the blue frame, which leads to the decoding bias about color information, i.e., "a black and white photo of a bicycle". Interestingly, the CMS of CLIP [CLS] token exhibits a significant discrepancy from the predictions made by the MindGPT. Since humans often pay attention to task-related objects (Shi et al., 2023), such visual cues appear to reflect human attention, motivating our future decoding efforts, i.e., attentional modulation-based reconstruction (Horikawa & Kamitani, 2022).

## 5 CONCLUSION

In this study, we have explored a non-invasive decoder when coupled with large (vision) language models to calculate the modality shift from visual to linguistic representations. Our initial results reveal that this simple, yet scalable, framework works surprisingly well, which suggests that there might be a rich connection between the amodal semantic concept and visual objects. While this hypothesis has been proposed in the neuroscience community, our study is the first to demonstrate that vision-to-language reasoning conditioned on a single brain image would be promising by using self-supervision models. A potential limitation is the accuracy ceiling imposed by pseudo-labels. Overall, our MindGPT is not only beneficial to decipher how the brain bridges different types of sensory information and then infers amodal semantic concepts, but also provides potential therapeutic values for people who are unable to communicate as a result of semantic dementia.

This work also leaves some open questions, and many challenges remain, although the potential of MindGPT is encouraging. One is that whether the amount of semantic information provided to the VC can be quantified by the selective visual attention of humans, which awaits further exploration and verification. Another question is how to explore the semantic relations between the VC and the anterior temporal lobe (ATL). The extensive evidence shows that ATL degeneration results in semantic dementia, and the answer to that question could help develop neuro-semantic prostheses for bypassing the ATL, thus recovering the loss of semantic signals due to ATL lesions.

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

# A APPENDIX

## A.1 NATURAL SCENES DATASET

The Natural Scenes Dataset (NSD) (Allen et al., 2022) is a recently released large-scale dataset that includes 7T fMRI responses to tens of thousands of natural scenes, with the goal of bridging cognitive neuroscience and AI. Specifically, NSD provides high-resolution fMRI recordings of eight subjects when they are presented with natural scenes, collected from the Common Objects in Context (COCO) dataset (Lin et al., 2014). During 30 to 40 MRI scanning sessions, each subject viewed between 9,000 to 10,000 different visual stimuli (with 22,000 to 30,000 repetitions).

## A.2 PERFORMANCE COMPARISON

To further validate the effectiveness of our MindGPT, we evaluate the reconstructed language in comparison with existing approach UniBrain (Mai & Zhang, 2023). The quantitative results on subject 1 of the NSD dataset are summarized in Tab. 3. Compared to ImageNet Deng et al. (2009), the COCO dataset provides more semantically complex scenes, and comprehensive annotations for both objects and scenes. That is, semantically complex scenes make it difficult to establish proper correspondences between virtual neural signals and candidate images. Consequently, in this context, we opt not to build a candidate image set for augmentation. We observe that MindGPT outperforms the competitor by 28%, 31%, and 1.8% respectively on ROUGE-1, ROUGE-L, and METEOR metrics.

| Model | Dataset | Subject | Language Similarity Metrics ↑ | | | | | |
|---|---|---|---|---|---|---|---|---|
| | | | B@1 | ROUGE-1 | ROUGE-L | METEOR | CIDEr | SPICE |
| UniBrain (Mai & Zhang, 2023) | NSD | sub-1 | - | 24.7 | 22.5 | 17.0 | - | - |
| MindGPT-L/16 (Ours) | NSD | sub-1 | 29.6 | **31.6** | **29.5** | **17.3** | 110.3 | 15.8 |

Table 3: Quantitative comparison of MindGPT and UniBrain on the NSD dataset (↑ denotes the higher the better). The results of UniBrain are taken from Mai & Zhang (2023), and the optimal indicator values are highlighted in **Bold**.

## A.3 IMPACT OF DATA AUGMENTATION

The final test was conducted to assess the impact of data augmentation techniques on MindGPT's performance. Tab. 4 summarizes the language decoding capabilities of our MindGPT under different numbers of candidate images. The ablation results demonstrate that data augmentation substantially improves MindGPT's decoding performance on the small-scale DIR dataset, and the quantity of candidate images also impacts the quality of the reconstructed word sequences.

| Model | Dataset | Subject | DA | Amount | Language Similarity Metrics ↑ | | | | | |
|---|---|---|---|---|---|---|---|---|---|---|
| | | | | | B@1 | B@4 | ROUGE-L | METEOR | CIDEr | SPICE |
| MindGPT-L/16 | DIR | sub-3 | × | - | 30.4 | 10.2 | 28.9 | 10.1 | 58.3 | 10.4 |
| | DIR | sub-3 | ✓ | 100 | 37.6 | 14.4 | 34.7 | 13.4 | 89.5 | 13.5 |
| | DIR | sub-3 | ✓ | 500 | 40.6 | 17.2 | 39.4 | 14.8 | 114.8 | 14.1 |
| | DIR | sub-3 | ✓ | All | **42.1** | **20.5** | **41.7** | **15.5** | **116.5** | **15.2** |

Table 4: Influence of the data augmentation (DA) on MindGPT. We report performance gains under varying numbers of candidate images. Note that the number of images here signifies the increase within the same category.

