# OpenReview forum: "MindGPT: Interpreting What You See with Non-invasive Brain Recordings"
_ICLR.cc/2024/Conference — Submitted to ICLR 2024_

### Official Review · Reviewer_QZRJ · 2023-10-27

**Soundness:** 3 good
**Presentation:** 1 poor
**Contribution:** 2 fair
**Rating:** 6
**Confidence:** 4

**Summary:**

The authors present an approach to decode semantic textual representations of images from brain activity data recorded with fMRI. An fMRI encoder (ViT) is trained (1) to predict the CLIP-Vision CLS embedding of the seen image given the fMRI activity and (2) as part of a pipeline where the CLIP prediction is fed to each layer of a frozen GPT-2 text generative model through cross-attention layers to predict the next word in the ground truth caption of the image. The pipeline is trained on the DIR dataset containing multiple presentations of 1200 training and 50 test images for three participants. Qualitative and quantitative results suggest the pipeline could predict similar captions to the ground truth. Further analysis showed that using voxels from the higher visual cortex areas leads to better reconstruction than lower or whole visual cortex areas.

**Strengths:**

Originality: The proposed pipeline (fMRI encoder + GPT-2 with cross-attention layers) appears like a novel application of a SMALLCAP-inspired approach (Ramos et al., 2023) but with CLIP latents predicted from fMRI and no retrieval-based prompting. However, similar fMRI-to-caption work has not been cited or compared to (see Weaknesses).

Quality: The paper is of acceptable quality, with a sound justification of the proposed approach and clear reporting of the results. However, as mentioned above, there is a lack of comparison to the existing literature.

Clarity: The paper is overall clear, though some information is missing (see Q1).

Significance: It is hard to evaluate the significance of the results given the lack of comparison to the existing literature. The analysis of Section 4.3 on decoding from specific ROIs provides interesting evidence into the properties of different cortical areas.

**Weaknesses:**

The main weakness to me is the lack of comparison to existing work on the topic of fMRI-to-caption decoding. I am aware of at least two papers proposing a similar fMRI-to-pretrained latent alignment + generative text modeling approach which also report qualitative and quantitative results on an fMRI-to-caption task (Mai & Zhang, 2023; Ferrante et al., 2023). However, the presented results are not compared to any previous baseline. Given the similarity of the approaches, a clear comparison must be made to establish whether the proposed approach provides an improvement over existing approaches. On a similar note, I believe a useful result to include in the analysis of Table 1 would be the performance of the pipeline if GPT-2 receives the ground-truth CLIP latents, instead of the fMRI-based predictions. This would provide an upper-bound for the proposed approach and help situate the reported results. Finally, the DIR dataset contains a small number of examples and categories as compared to the recent NSD dataset. As part of comparing the proposed approach to existing approaches, it would make sense to include results on this larger dataset as well.

Mai, Weijian, and Zhijun Zhang. "Unibrain: Unify image reconstruction and captioning all in one diffusion model from human brain activity." arXiv preprint arXiv:2308.07428 (2023).
Ferrante, Matteo, et al. "Brain Captioning: Decoding human brain activity into images and text." arXiv preprint arXiv:2305.11560 (2023).

**Questions:**

1. What is the value of H in the voxel vectors? Is there any lag between the image presentation and the extracted window? How much zero padding does this vectorization scheme lead to? Is there any aggregation of the presentations (at the BOLD or prediction level), or are the reported predictions obtained from single fMRI windows?

2. My understanding is that the input to the fMRI encoder is a sequence of 7 items (each one a vectorized set of ROI voxels) passed through a linear projection to 768 dimensions. Is the linear projection common to all items of the sequence? If so, I am curious to know what kind of operation it ended up learning to do.

3.  In Section 4.1: “Our MindGPT trained on DIR and a subset of ImageNet (Deng et al., 2009), including 150 categories totaling 200.7k images.” How was ImageNet used, and how were these 200.7k images selected? My understanding from Section 3.1 is that DIR contains 1250 unique images.

4. In Section 4.2: “Note that the default training/test split of DIR has no overlapping image categories, we randomly sampled 50 fMRI-image training pairs, and added them to the test set for the few-shot evaluation.” Can you confirm this means that these images were removed from the training set and added to the test set?

5. How do the results vary across subjects? It is not clear whether the results of Table 1 and 2 are across subjects or for a specific subject.

6. How were the examples of Figure 3 selected? What is the proportion of “failure cases” and high-quality reconstructions? Since there are only 50 test examples (times 3 subjects) my understanding is that all reconstructions could be presented in e.g. a table.

7. The analysis of Section 4.4 is an interesting way to see how the predicted latent shares information with the different image patches. What would this analysis give if you were to use the actual CLIP CLS token instead of the decoder’s prediction to compute the similarity scores? Would the scores look different for examples like the killer whale image (Figure 6, bottom left)? This might be a way to confirm that this analysis tells us about the brain decoding objective and not mostly the CLIP embedding itself.

---

> ### Author Response · Authors · 2023-11-22
>
> Thank you very much for your valuable feedback and suggestions.
>
> >1) **About the comparison.** Many thanks for pointing out the potential competitors for comparison. We did not compare with the approaches because they were not officially published nor provided source-code, at the time we submitted. Following your suggestion, we have added **comparative experiments (as well as a discussion) with UniBrain** [1], a recent preprint on arXiv, which demonstrated the strength and advantage of our approach (please refer to the Related Work and Appendix). We only included one subject currently due to time limitation and the experiment with all the subjects is running and will be added to the results. Many thanks.
>
> >2) **About the voxel vectors.** The value of H is 3745, determined by the maximum dimension among the voxel vectors, where the sizes of V1-V4, LOC, FFA, PPA are 2036, 2291, 2223, 1851, 3745, 980, and 908, respectively. The amount of zero padding equals H minus the size of each voxel vector.
>
> >3) **The details of MRI data preprocessing.** Data samples were created by first regressing out nuisance parameters from each voxel amplitude for each run. After that, voxel amplitudes were normalized relative to the mean amplitude of the initial 24s rest period of each run and were despiked to reduce extreme values. The voxel amplitudes were then averaged within each 8s (training) or 12s (test) stimulus block, after shifting the data by 4s to compensate for hemodynamic delays.
>
> >4) **About the few-shot reconstruction.** 150 training fMRI-image pairs across 150 classes were removed from the training set and added to the test set. At the stage of training, we did not provide any information about the correspondence between these signals and their stimuli.
>
> >5) **The results of Tabs 1 & 2.** In the initial-version paper, we use subject 3 of the DIR dataset. We have added **extra experiment on a large-scale fMRI dataset NSD** in the revised paper (please see Appendix).
>
> >6) **The role of linear projection.** Yes, the linear projection is shared across all items in the sequence (i.e., the input of the fMRI encoder). Due to the zero-padding operation and the low signal-to-noise ratio inherent in fMRI data, the input contains a considerable amount of redundant information. The linear projection of the fMRI encoder plays a role in dimensionality reduction, which helps the model focus more on important features and assists it in learning higher-level semantic representations, thus aiding the model in generalizing better to new, unseen data.
>
> >7) **Regarding the examples of Figure 3.** For the image selection in Fig. 3, we showed the images commonly used for qualitative comparison following the prior visual reconstruction studies, to compare with them intuitively. The ratio of high-quality reconstructions to failure cases is approximately 1:2.5.
>
> >8) **About the data augmentation.** The original DIR contains 1,250 images. We extracted all the images across 150 training classes from ImageNet as the candidate set for data augmentation. In other words, we relaxed the original one-to-one relationship to one-to-many. This practice not only provides more semantic supervision signals (image captions in the same category) for fMRI recordings, but also mitigates the effects of some pseudo labels with low confidence. We have added **an ablation experiment to evaluate the importance of data augmentations**.
>
> >9) **About the analysis of Section 4.4.** We appreciate your valuable and constructive advice. By utilizing the actual CLIP [CLS] token instead of the decoder’s prediction for computing similarity scores, we noted a significant discrepancy from the predictions made by the MindGPT. These findings have been integrated into the revised version.
>
> Many thanks for your constructive feedback, which significantly enhanced the overall quality of the paper. We have incorporated all the suggested modifications into the new manuscript. We hope our response can address your concerns. Would you please consider raising the scores?
>
> **References**
>
> [1] Mai, Weijian, and Zhijun Zhang. "Unibrain: Unify image reconstruction and captioning all in one diffusion model from human brain activity." arXiv preprint arXiv:2308.07428 (2023).

---

> > ### Comment · Reviewer_QZRJ · 2023-11-22
> >
> > Thank you to the authors for their answers and for providing a comparison with a baseline approach on NSD. This improves my confidence in the proposed approach. Based on this I increase my score to 6.
> >
> > Once results are obtained for all subjects, I would recommend moving these results to the main text, or at the very least mentioning them in the main text.

---

### Official Review · Reviewer_Hx5r · 2023-10-31

**Soundness:** 3 good
**Presentation:** 3 good
**Contribution:** 3 good
**Rating:** 5
**Confidence:** 4

**Summary:**

This paper introduces MindGPT, a non-invasive neural decoder that interprets perceived visual stimuli into natural languages from fMRI signals. The model employs a visually guided neural encoder with a cross-attention mechanism that uses the large language model GPT to guide latent neural representations toward a desired language semantic direction.

**Strengths:**

The proposed method, MindGPT, is a novel and innovative approach to interpreting visual stimuli using non-invasive brain recordings.
MindGPT has been shown to generate word sequences that truthfully represent the visual information conveyed in the seen stimuli, with essential details.
MindGPT has also been shown to be more semantically informative than other methods, and to be able to recover most of the semantic information using only the higher visual cortex (HVC).

**Weaknesses:**

About the novelty. Though the authors consider the proposed method as the first task-agnostic neural decoding model, CLIP-like models have been integrated in extensive research areas to make alignment with their specific representations in a similar way.
Also the evaluation, could you compare your model with exist CLIP-like models and show the strength of yours.
The dataset is not big enough to support the model.

**Questions:**

How well does MindGPT perform on a variety of different visual stimuli?
How does MindGPT compare to other state-of-the-art methods for interpreting visual stimuli using non-invasive brain recordings?
Could you add more detail on the dataset? How many subjects you used? How long the fMRI signal?
Does your work design a alignment between fMRI signal and images?

---

> ### Author Response · Authors · 2023-11-22
>
> Thank you very much for your valuable feedback and suggestions.
>
> >1) **About the novelty.** We proposed a novel idea to “mind reading” by interpreting brain signals into words. From the perspective of method, the data augmentation and cross-attention was the key technical component for attaining the goal, since the main challenge was how to extract the semantic component from limited fMRI data. In other words, our final goal is to guide latent neural representations towards a desired language semantic direction, instead of alignment with CLIP visual space. Unlike existing decoding paradigms [1-4], using simple linear regression models, our MindGPT is designed to explore self-supervised neural semantics learners by using cross-attention mechanisms and data augmentation techniques. We have modified our expressions in the paper to make the novelty clear. Many thanks.
>
> >2) **About the evaluation.** To best of our knowledge, there hasn't been any similar open-source work on brain-to-text tasks published in peer-reviewed journals or conferences. To further validate the effectiveness of our approach, **we have included comparative experiments with UniBrain [3] (using CLIP model)**, which is a recent preprint available on arXiv (please see Appendix).
>
> >3) **About the details of this work.** **a) Subject**: In the initial-version paper, we use subject 3 of the DIR dataset. **We have added extra experiment on a large-scale fMRI dataset NSD in the revised paper** (please see Appendix); **b) Alignment**: During the training process, we applied a CLIP-based L2 constraint to regulate the fMRI embeddings; **c) fMRI data preprocessing**: Data samples were created by first regressing out nuisance parameters from each voxel amplitude for each run. After that, voxel amplitudes were normalized relative to the mean amplitude of the initial 24s rest period of each run and were despiked to reduce extreme values. The voxel amplitudes were then averaged within each 8s (training) or 12s (test) stimulus block, after shifting the data by 4s to compensate for hemodynamic delays.
>
> Thanks for the constructive comments, which helped improve the paper’s quality. We have incorporated all the suggested modifications into the new manuscript. We sincerely hope our response can address most of your concerns. Would you please consider raising the scores?
>
> **References**
>
> [1] Matsuo, Eri, et al. "Describing semantic representations of brain activity evoked by visual stimuli." 2018 IEEE International Conference on Systems, Man, and Cybernetics (SMC). IEEE, 2018.
>
> [2] Takada, Saya, et al. "Generation of viewed image captions from human brain activity via unsupervised text latent space." 2020 IEEE International Conference on Image Processing (ICIP). IEEE, 2020.
>
> [3] Mai, Weijian, and Zhijun Zhang. "Unibrain: Unify image reconstruction and captioning all in one diffusion model from human brain activity." arXiv preprint arXiv:2308.07428 (2023).
>
> [4] Ferrante, Matteo, et al. "Brain Captioning: Decoding human brain activity into images and text." arXiv preprint arXiv:2305.11560 (2023).

---

### Official Review · Reviewer_SjKZ · 2023-11-01

**Soundness:** 4 excellent
**Presentation:** 3 good
**Contribution:** 2 fair
**Rating:** 6
**Confidence:** 3

**Summary:**

In this paper authors propose MindGPT to generate captions of the images perceived by humans from the fMRI responses during perception. To do so first they align fMRI responses to visual feature space (CLIP) using a fMRI encoder (ViT) and then guide a language decoder to generate captions of images from fMRI embeddings using cross attention.

The training and evaluation is performed on publicly available fMRI dataset(Horikawa and Kamitani, 2017 ; Shen et al. ). The results show that captions generated by MindGPT correctly capture some of the semantic information present in the scene perceived. They also perform additional analysis which show

1. voxels from higher visual cortex (HVC) lead to more accurate captions as compared to lower visual cortex (LVC).
2. which visual cues were relevant for caption generation

**Strengths:**

1. The paper is easy to follow, well written with descriptive figures.
2. Although the idea to generate captions from fMRI responses is not new and has been explored previously (see references in weaknesses section) but the use of recent methods such as image captioning (SMALLCAP) to generate pseudo groundtruth captions, alignment with CLIP encoder and use of cross attention to guide GPT-2 makes this a new contribution
3. Qualitatively the results are impressive (I would have preferred to see the captions of all 50 test images ). I do not have expertise in captioning literature so I am not sure how good are quantitative results though.
4. The analysis showing HVC generates more accurate captions is exciting and could be combined with reconstruction methods to generate reconstructions that are both pixel-level and semantically accurate. In reconstruction LVC contribution is more, this paper shows that complementary information can be decoded from HVC.
5. Authors perform additional analysis to show which visual cues were relevant for caption generation and tsne analysis to show latent representation of different brain regions
6. Use of virtual training examples to augment the smaller training dataset.

**Weaknesses:**

1. The results in this paper are from DIR dataset (1200 training and 50 test images) which is smaller compared to NSD dataset (10k images). Another advantage of NSD is that images are from MS-COCO dataset which contain more semantically complex scenes as compared to imagenet images in DIR which contain a single object centered. So, I would like to know author’s reasoning on why NSD dataset was not considered for this paper.
2. There are a few relevant references missing (Matsuo et al. , Sakata et al.) which generate captions from brain activity. I am not sure if the code of these papers are available that’s why I am not asking to compare the results but I believe they should be at least discussed for readers to understand how this paper is different from previous fMRI→ caption generation methods.
3. Self-attention maps of MindGPT encoder can be used to inform which brain region was more relevant in generating captions. This analysis could have provided more insight into captions of which category of images are generated by which brain regions.
4. It is not clear what groundtruth is used to compute language similarity metrics. If it is compared to Image captions generated by SMALLCAP then it is a major limitation of this approach. Accuracy of image generated captions will be upper bound by SMALLCAP. Collecting human captions for at least the test set and comparing both SMALLCAP and MindGPT would have been more informative.
5. Minor: I assume blue color text in Figure 3,4 indicates correct captions and black colored text indicate incorrect text. These should be mentioned in the Figure captions for clarity.

- Matsuo, Eri, et al. "Describing semantic representations of brain activity evoked by visual stimuli." *2018 IEEE International Conference on Systems, Man, and Cybernetics (SMC)*. IEEE, 2018.
- Takada, Saya, et al. "Generation of viewed image captions from human brain activity via unsupervised text latent space." *2020 IEEE International Conference on Image Processing (ICIP)*. IEEE, 2020.

**Questions:**

Questions

1. Why was NSD not considered for this paper?
2. Was any ablation performed to assess how crucial was the data augmentation performed using virtual training examples? e.g. will including more images per category further improve results?

Clarification

1. Identification of visual cues using cosine similarity between fMRI encoder output and patch tokens inform which cues can be extracted from fMRI responses? Can this be helpful in answering where in an image subject was attending. If that is the case generating captions from fMRI data of Horikawa and Kamitani 2022 can lead to interesting findings.

Suggestions
1. Please refer to weaknesses section. I am confident that this contribution has the potential to be a good paper if the authors address weaknesses.

Reference:
Horikawa, Tomoyasu, and Yukiyasu Kamitani. "Attention modulates neural representation to render reconstructions according to subjective appearance." *Communications Biology* 5.1 (2022): 34.

---

> ### Author Response · Authors · 2023-11-22
>
> Thank you very much for your valuable feedback and suggestions.
>
> > 1) **Regarding the NSD dataset.** The reason we did not use the NSD was that the MS-COCO dataset contains detailed annotations for images, but it does not provide specific classification labels like ImageNet. In other words, semantically complex scenes make it difficult for us to establish proper correspondences between virtual neural signals and candidate images. On the contrary, the images of ImageNet (typically including a single object) allow us to exploit virtual fMRI-image correspondences under the same class semantics.  We do agree that NSD dataset has advantages over DIR dataset in terms of the scale (10k images) and semantics richness. Therefore, we have added **an extra experiment with NSD dataset in comparison of a preprint work [3] (also used NSD)**. We only included one subject currently (please see Appendix) due to time limitation and the experiment with all the subjects is running and will be added to the results. Many thanks.
>
> >2) **About the ablation study.** We have added an ablation experiment to evaluate the importance of data augmentation, which has been incorporated in the Appendix of the revised paper. Many thanks.
>
> >3) **About the quantitative results.** The results in the paper are obtained via calculating language similarity metrics between the pseudo-caption (provided by SMALLCAP) and the generated text of MindGPT. We agree that this is a potential limitation of the approach. During the training phase, we used 200.7k candidate images for augmentation, making it challenging to collect human captions, but we are working in that direction.
>
> >4) **The discussion about relevant references.** Previous works [1-4] mapped fMRI recordings to the embeddings of pre-trained neural networks like VGGNet by relatively simple linear regression models, and then feeds the predicted result into language models to generate word sequences. Unlike existing decoding paradigms, our MindGPT is designed to explore self-supervised neural semantics learners by using cross-attention mechanisms and data augmentation techniques. To the best of our knowledge, generating linguistic semantic information directly from a single brain image in an end-to-end fashion has not been adequately explored. The above discussion has been incorporated into the revised version. Many thanks.
>
> >5) **About the visual cues.** The visual cues demonstrated that the model autonomously acquired information about the salient regions within an image, a process devoid of any explicit labels or supervision. By utilizing the actual CLIP [CLS] token for computing similarity scores, we noted a significant discrepancy from the predictions made by the MindGPT. These findings have been integrated into the revised version. Our preliminary hypothesis is that the visual cues are related to the subjects' eye movements when viewing the images. Comparing them (i.e., attention maps) to eye movement data may provide a more convincing answer. We think it would be a valuable research topic in our future studies.
>
> >6) **About the weakness.** The discussion of the limitations, e.g., the upper bound of accuracy, of this work has been incorporated into the revised version. Many thanks.
>
> >7) **The Captions of Figs 3, and 4.** Blue color text in Figs 3, and 4 indicates correct (or semantically similar) captions and incorrect text is highlighted in red. We have added explanations in captions of Figs 3, and 4.
>
> Thanks for the valuable comments, which we do believe help improve the accessibility of the paper. We have incorporated all the suggested modifications into the resubmitted manuscript. We sincerely hope our response can address your concerns. Would you please consider raising the scores?
>
> **References**
>
> [1] Matsuo, Eri, et al. "Describing semantic representations of brain activity evoked by visual stimuli." 2018 IEEE International Conference on Systems, Man, and Cybernetics (SMC). IEEE, 2018.
>
> [2] Takada, Saya, et al. "Generation of viewed image captions from human brain activity via unsupervised text latent space." 2020 IEEE International Conference on Image Processing (ICIP). IEEE, 2020.
>
> [3] Mai, Weijian, and Zhijun Zhang. "Unibrain: Unify image reconstruction and captioning all in one diffusion model from human brain activity." arXiv preprint arXiv:2308.07428 (2023).
>
> [4] Ferrante, Matteo, et al. "Brain Captioning: Decoding human brain activity into images and text." arXiv preprint arXiv:2305.11560 (2023).

---

> > ### Comment · Reviewer_SjKZ · 2023-11-22
> >
> > Thanks for addressing my concerns and adding new results. I find this paper has improved in revision and therefore have increased my rating

---

### Official Review · Reviewer_7fxY · 2023-11-01

**Soundness:** 4 excellent
**Presentation:** 3 good
**Contribution:** 3 good
**Rating:** 6
**Confidence:** 3

**Summary:**

This paper proposes a new method to reconstruct visual stimuli from the brain activities using fmri, that seeks to decode the information encoded in the human brain when processing visual information. One of the main bottlenecks of this approach is the limit on acquiring more high quality features while staying non invasive. Previous work has achieved only limited success in this field, where most of the reconstructions are blurry and without any low-level texture in the image. Having this said, the authors open new doors by using semantic feature extraction as opposed to pixel wise, feeding them to another generative model. They argue that the mentioned method is not only more robust, but obtain more detailed and relevant images after all.

**Strengths:**

1. Paper is very well-written even for a general reviewer. Furthermore, the settings of the model and experiments are accurately described as to make it easier for reproducibility.
2. The novelty of using semantic features in the middle of their end-to-end model is quite intuitive. Plus this message has been delivered quite clear to the reader.
3. Using augmentation to compensate that lack of data is justified, and they modify already existing methods to apply better in their case.
4. Formulation of the loss is carefully written to adhere to the main purpose of the paper.
5. Images generated with the proposed algorithm contain more low-level details and texture as shown in the paper.

**Weaknesses:**

1. The paper has circumvented the main goal of visual reconstruction, we are completely relying on the generative model to build the images.  In other words, images generated this way won't be closer than a certain amount despite having enormous details.
2. Comparison with other methods in the field has been mentioned scarcely after the introduction part, e.g., in the experiments it's done only among the different versions of their own algorithm.

**Questions:**

Can you please mention more methods before this paper that aimed the same goal? Also highlight if using semantic methods are a complete novelty or it has been used before to some extent, with other approaches.

---

> ### Author Response · Authors · 2023-11-22
>
> Thank you very much for your valuable feedback and suggestions.
>
> >1)  **About visual reconstruction.** Our goal is to interpret the scene we see rather than reconstruct the image. The underlying hypothesis is that our brain is not a camera, and we can’t actually store all the pixels we see. Instead, we process visual information (at a semantic level) in our brain. Thus, we think that decoding semantic information and expressing it linguistically could be a more efficient way to understand the brain.
>
> >2) **The discussion and comparison with other competitors.** Decoding semantic sentences instead of reconstructing the image is a novel perspective and to best of our knowledge, there hasn’t been any similar open-source study officially published at the time we submitted. To validate the effectiveness of our approach, we have added **comparative experiments (as well as a discussion)** with UniBrain [1], a recent preprint on arXiv, which demonstrated the strength and advantage of our approach (please refer to the Related Work and Appendix).
>
> We sincerely hope our response can address most of your concerns. Would you please consider raising the scores?
>
> **References**
>
> [1] Mai, Weijian, and Zhijun Zhang. "Unibrain: Unify image reconstruction and captioning all in one diffusion model from human brain activity." arXiv preprint arXiv:2308.07428 (2023).

---

### Meta-Review · Area_Chair_qiBQ · 2023-12-10

**Metareview:**

The paper aims to reconstruct static images from fMRI leveraging a pretrained LLM (GPT-2) and image embedding. One original ideal is to use the LLM to generate captions which are exploited as text to condition the image generation.

The paper is judged by all reviewers as well written with convincing experimental results.

The use of the captioning dataset is original.

There are however some concerns about how well the method compares with previous approaches aiming to reconstruct images from fMRI data.

**Justification For Why Not Higher Score:**

The ML contribution is not strong and the results are not clearly beyond what has been proposed before.

**Justification For Why Not Lower Score:**

N/A

---

### Decision · Program_Chairs · 2024-01-16

Reject